# A Kinematic Collision Box Algorithm Applied for the Anti-Collision System of Offshore Drilling Vessels

**Duy Thanh Nguyen [1], Kwang Hyo Jung [1], Ki-Youn Kwon [2], Namkug Ku [3] and Jaeyong Lee [3,\*]** 

1   Department of Naval Architecture and Ocean Engineering, Pusan National University, Busan 46241, Korea; duythanhnguyen@pusan.ac.kr (D.T.N.); kjung@pusan.ac.kr (K.H.J.)
2   School of Industrial Engineering, Kumoh National Institute of Technology, Gyeongbuk 39177, Korea; mrkky@kumoh.ac.kr
3   Department of Naval Architecture and Ocean Engineering, Dong-eui University, Busan 47340, Korea; knk80@deu.ac.kr
\*   Correspondence: jlee@deu.ac.kr; Tel.: +82-51-890-2596

**Abstract:** With the advances in technology and the automation of drilling platforms, the Anti-Collision System (ACS) has appeared as an affordable technology, which is intended to keep equipment on the drilling floor working harmoniously and to prevent the potential hazards associated with accidents. However, the specialty of the machinery on the drilling floor requires a distinguished structure for the ACS and a reliable collision-avoidance algorithm, which is not similar to any algorithm in other applications, such as automobiles and robotics. The aim of this paper is to provide a comprehension of the configuration of an ACS in an Integrated Drilling System and to develop a practical anti-collision algorithm that can be applied to the machine arrangement for an offshore drilling operation. By analyzing the motions and using kinematic parameters, such as the speed and deceleration information of drilling equipment, a kinematic collision box algorithm is developed to eliminate the limitation of conventional algorithms. While the conventional collision-avoidance algorithm uses a collision box with fixed size, the kinematic collision box algorithm uses a collision box with a flexible scale that can be correspond to the velocity and deceleration rate of the equipment. Several operating scenarios are simulated by a visual model of ACS to authenticate the functionality of the proposed algorithm. The operation of the top drive is an outstanding scenario. Only 2.25 s are required to stop the top drive from its maximum velocity, and a conventional algorithm uses this number to create a fixed bounding box. Also, the kinematic collision box algorithm uses the real-time data of velocity and acceleration to adjust the scale of the bounding box when the speed of the top drive increases from 0 to its maximum value. The simulation result illustrates the reliability and advances of the kinematic collision box algorithm in performing the collision-avoidance function in ACS compared to the conventional algorithm.

**Keywords:** Anti-Collision System; Integrated Drilling System; offshore drilling floor; collision check algorithm; kinematics; bounding box; collision box

## 1. Introduction

The upstream sector of the oil and gas industry has undergone significant changes as the technology has evolved with time. Besides the problem of optimizing the efficiency, safety is always a predominate concern in a petroleum field, particularly during drilling activities. The process of drilling an oil well requires extensive movements of equipment and numerous operations, both of which occur simultaneously at high frequency. The drilling process of an offshore vessel is more complicated and higher chance of accidents if the vessel has a dual derrick system.

The development of technology has provided several solutions to optimize the drilling process and to minimize the possibility of accidents [1]. Continued improvements and the automation of the process have reduced the amount of manual work required since most of the equipment is controlled by computers or by commands from personnel stationed at a remote control system [2].

A benefit of using automation technology on the drilling floor is that equipment can perform a variety of tasks automatically, which is especially efficient for operations that consist of repetitive tasks such as tripping and stand-building processes [2]. However, limitations in the drilling space, where equipment is operated and potential failures in the control system can result in collisions or other interferences between the equipment. Thus, a tool that is integrated with drilling control system is required to prevent potential collisions and other interferences.

The acronym, ACS, has been used to represent the words Anti-Collision System for a long time in many fields, including automation, robotics, and marine transportation, but there several inhibitions to its use in reference to the in the drilling floor [3–5]. The term ACS has appeared in the petroleum industry, but it generally indicates a method to prevent the collision of wellbores at subsurface area, not the collision between drilling equipment on the drilling floor [6].

ACS must ensure that the automated equipment works harmoniously, and it must perform the task in a minimum amount of time. The development of computer science has offered efficient methods that can be used to create a reliable and optimized ACS for drilling machines [7].

One of the feasible methods is to visualize the drilling floor and related equipment in a computer environment and apply a collision detection algorithm to prevent interferences between the various kinds of equipment. However, the drilling operation includes numerous tasks, and the movements of equipment are quite complicated. Currently, ACS applied in real-life drilling systems are using collision boxes with fixed scales, which is inappropriate for the various rates of kinematic parameters of machinery on the drill floor. Thus, the development of a kinematic collision box algorithm is an essential feature to implement the ACS to real-life systems. ACS for offshore drilling floors require carefully designed algorithms, which must be reliable and have ability to perform the task in a minor interval of time.

The remainder of this paper is structured as follows: Section 2 describes the configuration of the ACS as a part of an Integrated Drilling System, and the processing sequence of the whole system. Conventional collision check algorithm and the proposed kinematic collision box algorithm are presented in Section 3, and some simulations for the drill floor in an offshore vessel are conducted and the results are compared in Section 4. Finally, a brief conclusion is made in Section 5.

## 2. Overall Architecture of the ACS

### 2.1. Configuration of the ACS in an Integrated Drilling System

2.1.1. Integrated Drilling System

Integrated Drilling Systems (IDSs) are based on a computer network that interconnects all the drilling machinery and sensors based on a distributed supervisory and control concept. This approach is used to implement automated operations and full remote control from a central control room, and multimedia technology is used for the man-machine interface [8].

The implementation of IDSs allows complex and dangerous tasks to be performed by machinery in either a fully automatic mode or semi-automatic mode controlled by operators stationed in a remote control room. In addition, the drilling process is monitored closely through IDS because information can be shared between various subsystems, and all parties can be involved in the drilling process through a communication network and a real-time database [9]. Figure 1 shows the configuration of the IDS.

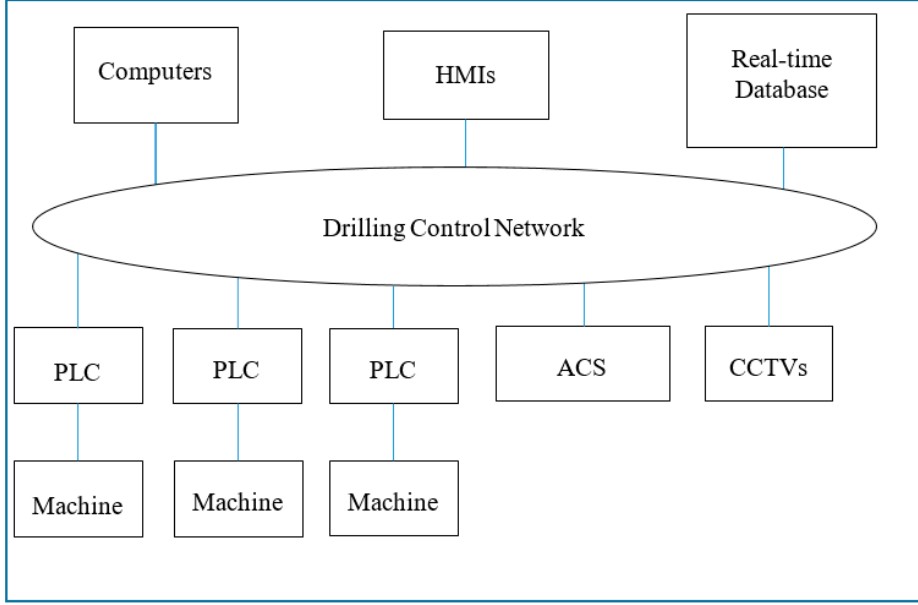

**Figure 1.** Structure of the Integrated Drilling System.

The structure of the IDS includes the following components:

a.  Drilling Control Network (DCN)
b.  Programmable Logic Controllers (PLCs)
c.  Computers
d.  A real-time database
e.  Human–Machine Interfaces (HMIs)
f.  Auxiliary tools [8]

The IDS is designed as an open computer platform, so the integration of auxiliary software to support the drilling operation is totally feasible and ACS is a practical example.

In IDS, computers are in the upper level of the hierarchy, which have the role to supervise subsystem and perform analyzing tasks. HMIs help users to monitor the operation by visualizing data from the system and providing necessary control functions to subsystems. PLCs, with their associated machinery and I/O equipment are at lower level of IDS and perform different tasks that are required in the overall process.

All information in the system is processed and exchanged via a DCN and a real-time database. The progress of the IDS provides the ability to upgrade the drilling platform with up-to-date tools and software, such as ACS, which makes the operation safer and more efficient [10].

### 2.1.2. Components of ACS

ACS monitors equipment that moves equipment on a drilling rig, including traveling blocks, top drive, bails/elevator, and additional pipe handling systems. The speed of the traveling block can be controlled by a 'soft stop' to prevent collisions between moving equipment, floor, and derrick structure. ACS monitors all drilling floor equipment, keeping the machines on a safe distance from each other. This keeps the personnel safe, prevents damage to equipment, and enhances uptime on the rig [11].

The configuration of ACS includes the following components:

a.  Anti-Collision Software: to determine stop positions and stop commands by using a collision detection algorithm.
b.  HMI: to represent the status of equipment, the ACS matrix, and messages.
c.  ACS Viewer: to provide 3D visualization of drilling floor and bounding boxes of equipment.
d.  Anti-Collision Bypass Control Module: override function switches.

### 2.1.3. Network Topology between ACS and DCN

The communication between ACS and the drilling system is achieved by the network topology of IDS (IP/F or TCP/IP Protocol), which has a ring-type construction. Figure 2 illustrates the communication network in an IDS. Through the network topology, ACS receives necessary data, including positions, velocity, deceleration, and other relevant data related to equipment to analyze the motions of equipment in the coordinate of the drilling floor. After using the algorithm to check for potential collisions by the algorithm, ACS will return feedback signals concerning the results of the collision detection process, and it will issue warnings, bypass control signals, or stop commands to the server. In addition, ACS uses the result of the collision-checking process to update the functionality of the Viewer.

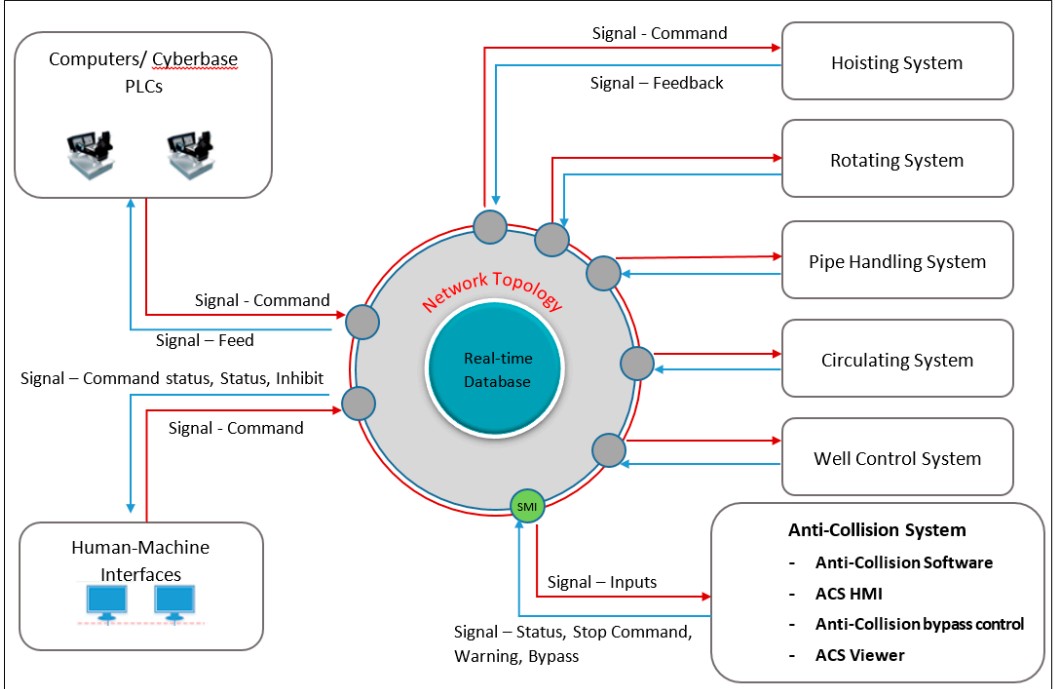

**Figure 2.** Communication Network in an Integrated Drilling System.

### 2.2. ACS Software

ACS software receives data from IDS in the bit chain format, which includes machine ID, position of the machine. The network module and the data conversion module of the software have the responsibility to receive and derive data from an array into separated variables. The software uses this information to update the properties of objects then check with override function to determine whether there is any signal of Ignore or Release. The collision-checking function is performed by an algorithm, which is initialized based on the C++ language and included in the software. The result of collision-checking is analyzed to generate signals and messages and to transfer to machines' PLCs through the network. In addition, the ACS viewer is updated based on the results of collision-checking algorithm.

In this research, Unreal Engine 4 (Version 22.2, developed by Epic Games) is designated to perform the collision detection task in the ACS because it offers an environment to visualize 3D objects with complicated movements, and it also has integrated functions that can check the collisions between several objects. Figure 3 shows the data processing sequence in ACS hosted by Unreal Engine 4.

Unreal Engine is a software program with real-time technology functions that normally are used in developing games [12]. This program also provides high-quality, photorealistic renders and immersive augmented reality (AR) and virtual reality (VR) experiences for architecture, automotive, film and television, training and simulation, and other industries other than games [12].

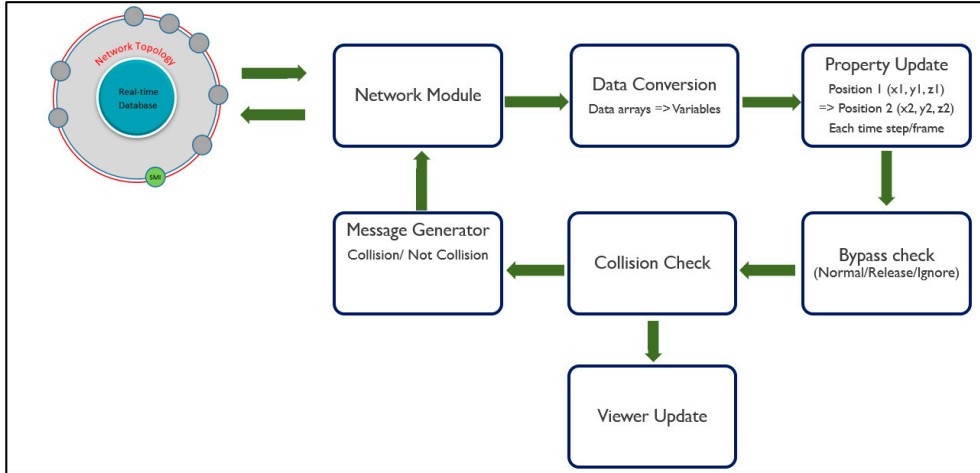

**Figure 3.** Data processing sequence in ACS hosted by Unreal Engine 4.

## 3. Collision Check Algorithm

### 3.1. Conventional Collision Box

#### 3.1.1. Collision Detection

The process of determining whether two or more bodies are making contact at one or more points is called collision detection or intersection detection [13]. Collision detection is an inseparable part of computer graphics, surgical simulations, and robotics. There are several methods for detecting collisions, including Spatial Partitioning Representations, Octree, Bounding Box Volume Hierarchies, and Binary Space Partitioning Tree [14].

The most common method is based on the Bounding Box Volume Hierarchies. The bounding volume of an object has variety geometry from simple shapes, such as sphere, cube, and cylinder to complex polygons. The major reason for the popularity of the Bounding Box Volume Hierarchies method is that it uses a bounding volume that usually is the simplest geometric primitive that encloses one or more objects to conduct overlapping tests. It is considered to be a cheaper test, but still has high efficiency when compared with other methods [15].

After considering the design and motion of machinery on the drilling floor, the axis-aligned bounding box and the oriented bounding box offers accordant because of the following factors [16]:

a.   They provide an inexpensive test for intersection.
b.   They fit tightly around the bounded object(s).
c.   They are inexpensive to compute.
d.   They can be easy transformed (e.g., rotated).
e.   They require little memory.

The disadvantage of applying this method to the ACS on drilling floor is that the information about the position and kinematic parameters of the equipment need to be updated to the algorithm continuously. Thus, it requires the drilling system to be designed with modern technology, which allows the information to be acquired and transferred accurately

#### 3.1.2. Axis-Aligned Bounding Box (AABB) and Oriented Bounding Box (OBB) Collision Detection

An AABB is a rectangular box whose face normals are parallel to the axes of the coordinate system [16]. An AABB can be represented compactly as a center point and the associated distances of the half-width axis-extents [17]. Figure 4 shows the examples of AABB and OBB of an equipment on the drilling floor. For instance, if box A is defined by $x_{minA}$, $x_{maxA}$, $y_{minA}$, $y_{maxA}$, $z_{minA}$, and $z_{maxA}$ and

box B is defined by $x_{minB}$, $x_{maxB}$, $y_{minB}$, $y_{maxB}$, $z_{minB}$, and $z_{maxB}$, then the boxes overlap if all of the following conditions are true [18]:

$$x_{minA} < x_{maxB} \text{ and } x_{maxA} > x_{minB}$$
$$y_{minA} < y_{maxB} \text{ and } y_{maxA} > y_{minB} \tag{1}$$
$$z_{minA} < z_{maxB} \text{ and } z_{maxA} > z_{minB}$$

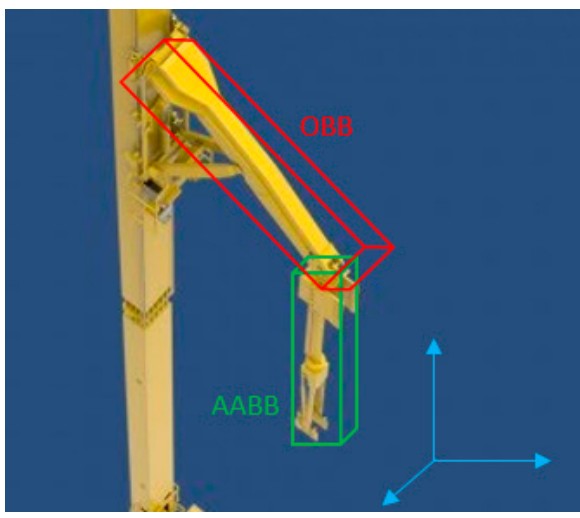

**Figure 4.** Axis-aligned bounding box and oriented bounding box of objects.

An OBB is an arbitrarily oriented rectangular bound. An OBB is simply a bounding parallelepiped whose faces and edges are not parallel to the basis vectors of the frame in which they are defined. One popular way to define them is to specify a center point C and orthonormal set of basis vectors {*i, j, k*}, which determines location and orientation, and three scalars representing the half-width, half-length, and half-height [19]. Intersection testing is more complex than testing for an AABB and a separating axis test is often used [20].

The Separating Axis Theorem (SAT) is used to determine if two arbitrary shapes intersect. Both shapes that are tested must be convex. The SAT works by looking for at least one axis of separation between two objects. If no axis of separation exists, the objects are colliding [21]. Figure 5 shows a example of using SAT to check the collision between 2 objects.

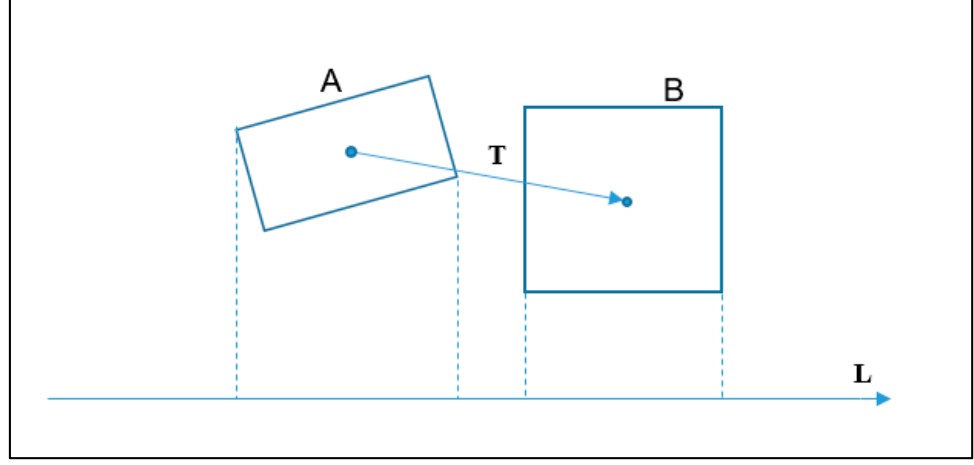

**Figure 5.** Separating Axis Theorem in 2D space.

Let A and B be OBB defined by:

$P_A$ = coordinate position of the center of A

$i_A$ = unit vector representing the local x-axis of A

$j_A$ = unit vector representing the local y-axis of A

$k_A$ = unit vector representing the local z-axis of A

$W_A$ = half-width of A (corresponds with the local x-axis of A)

$L_A$ = half-length of A (corresponds with the local y-axis of A)

$H_A$ = half-height of A (corresponds with the local z-axis of A)

$P_B$ = coordinate position of the center of B

$i_B$ = unit vector representing the local x-axis of B

$j_B$ = unit vector representing the local y-axis of B

$k_B$ = unit vector representing the local z-axis of B $W_B$ = half-width of B (corresponds with the local x-axis of B) $L_B$ = half-length of B (corresponds with the local y-axis of B) $H_B$ = half-height of B (corresponds with the local z-axis of B)

Variable

$T = P_B - P_A$: vector

Here is the general inequality:

$$\left| T \cdot L \right| > \left| (W_A i_A) \cdot L \right| + \left| (L_A j_A) \cdot L \right| + \left| (H_A k_A) \cdot L \right| + \left| (W_B i_B) \cdot L \right| + \left| (L_B j_B) \cdot L \right| + \left| (H_B k_B) \cdot L \right| \tag{2}$$

There are 15 separating axes that need to be inspected for 2 boxes in 3D space corresponds to 15 different vector $L$.

## 3.2. Kinematic Collision Box

### 3.2.1. Kinematic Collison Box Algorithm for Drilling Floor

The anti-collision mechanism of ACS is based on bounding box volume collision detection, and these algorithms and Unreal Engine already have included these collision detection algorithms. However, the bounding box of an object that is self-constructed by Unreal Engine is the smallest cube that encloses the object. Thus, Unreal Engine will trigger the interferences only when objects collided together. However, the function of ACS is to prevent the collisions and it must determine potential interferences and stop the machine before collisions occur.

In this paper, the term "kinematic collision box" algorithm indicates an algorithm that uses the bounding box of an object in Unreal Engine to create an anti-collision box of an object that will be applied in ACS. The algorithm will rely on the motion vector, velocity, acceleration, and deceleration rate of the object to determine the swept volume of AABB or the stop position of the OBB bounding box. Figure 6 illustrates the function of the kinematic collision box algorithm in the collision detection sequence.

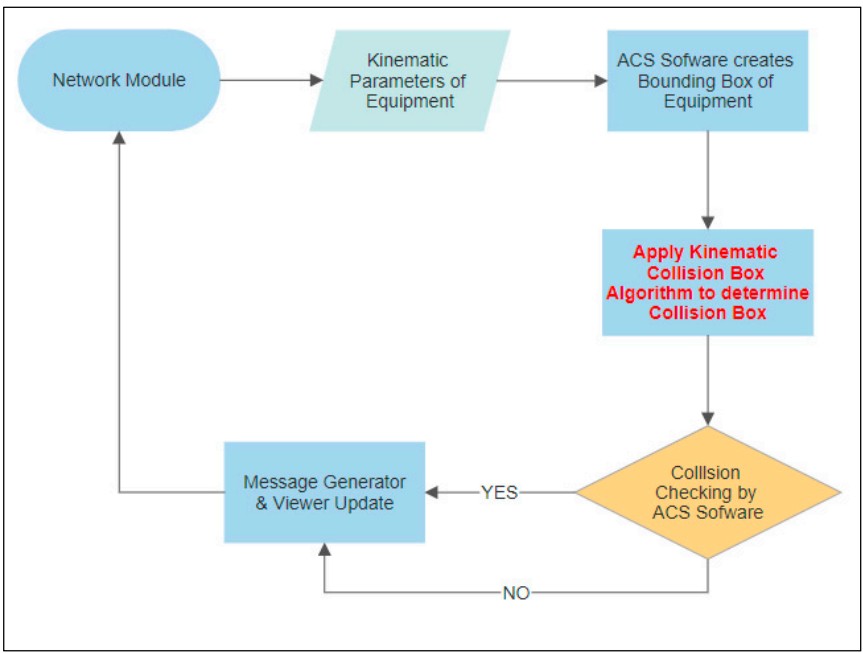

**Figure 6.** Kinematic Collision Box Algorithm in Data Processing Sequence.

3.2.2. Axis-Aligned Kinematic Collision Box

An object A with the AABB is defined by:

A $(x_A, y_A, z_A)$ = coordinate position of the center of A
$W_A$ = half-width of A (corresponds with the local x-axis of A)
$L_A$ = half-length of A (corresponds with the local y-axis of A)
$H_A$ = half-height of A (corresponds with the local z-axis of A)

Figure 7 illustrates an example of Axis-Aligned Kinematic Collision Box algorithm. If object A is moving along 3 axes X, Y, and Z with velocities $v_x$, $v_y$, and $v_z$, and minimum decelerations $a_x$, $a_y$, and $a_z$ corresponds with 3 local axes X, Y, and Z, the axis-aligned kinematic collision box of object A in the general case would be defined as:

$x_A + \frac{1}{2}\left(v_x t_x - \frac{1}{2}a_x t_x^2\right); y_A + \frac{1}{2}\left(v_y t_y - \frac{1}{2}a_y t_y^2\right); z_A + \frac{1}{2}\left(v_z t_z - \frac{1}{2}a_z t_z^2\right)$ : Coordinate position of the center of kinematic collision box of A, with $t_x = v_x/a_x$; $t_y = v_y/a_y$; $t_z = v_z/a_z$.

$W_C = W_A + \left|\frac{1}{2}\left(v_x t_x - \frac{1}{2}a_x t_x^2\right)\right|$: Half-width of Axis-Aligned Kinematic Collision Box of A

$L_C = L_A + \left|\frac{1}{2}\left(v_y t_y - \frac{1}{2}a_y t_y^2\right)\right|$: Half-length of Axis-Aligned Kinematic Collision Box of A

$H_C = H_A + \left|\frac{1}{2}\left(v_z t_z - \frac{1}{2}a_z t_z^2\right)\right|$: Half-height of Axis-Aligned Kinematic Collision Box of A

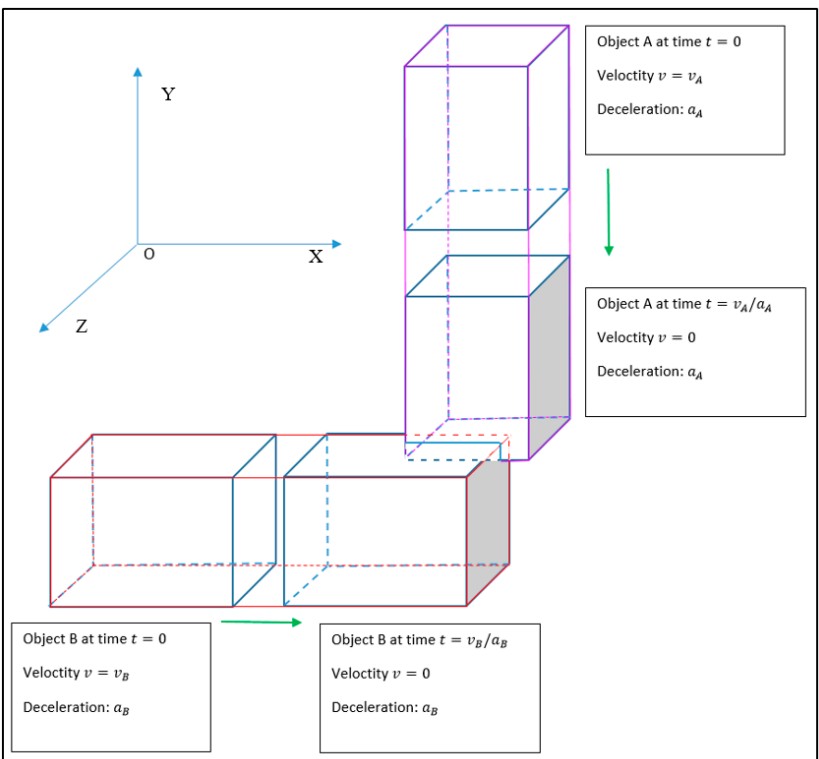

**Figure 7.** Axis-aligned anti-collision box of objects.

### 3.2.3. Oriented Kinematic Collision Box

With objects that contains rotating movement, the axis-aligned kinematic collision box is invalid after the transformation of the object. Therefore, the oriented kinematic collision box is applied, and it is determined by the maximum time required to stop an object that is moving with both linear and angular velocities. Figure 8 is an example of oriented kinematic collision box algorithm.

An object A with the OBB, which is defined by:

A $(x_A, y_A, z_A)$ = coordinate position of the center of A
$i_A$ = unit vector representing the local x-axis of A
$j_A$ = unit vector representing the local y-axis of A
$k_A$ = unit vector representing the local z-axis of A
$W_A$ = half-width of A (corresponds with the local x-axis of A)
$L_A$ = half-length of A (corresponds with the local y-axis of A)
$H_A$ = half-height of A (corresponds with the local z-axis of A)
$\varphi_A, \theta_A, \psi_A$: Three Euler angles to determine the orientation of bounding box

In a general case, the oriented kinematic collision box would be defined as:

$$x_A + \left(v_x t_x - \tfrac{1}{2}a_x t_x{}^2\right); \; y_A + \left(v_y t_y - \tfrac{1}{2}a_y t_y{}^2\right); \; z_A + \left(v_z t_z - \tfrac{1}{2}a_z t_z{}^2\right):$$

Coordinate position of the center of oriented anti-collision box of A, with $t_x = v_x/a_x$; $t_y = v_y/a_y$; $t_z = v_z/a_z$

The Euler angles of the oriented kinematic collision box of A are as follows:

$$\varphi_C = \varphi_A + \left| \tfrac{1}{2}\left( \omega_x t_\varphi - \tfrac{1}{2}\alpha_x t_\varphi{}^2 \right) \right|; \ t_\varphi = \omega_x/\alpha_x;$$

$$\theta_C = \theta_A + \left| \tfrac{1}{2}\left( \omega_y t_\theta - \tfrac{1}{2}\alpha_y t_\theta{}^2 \right) \right|; \ t_\theta = \omega_y/\alpha_y;$$

$$\psi_C = \psi_A + \left| \tfrac{1}{2}\left( \omega_z t_\psi - \tfrac{1}{2}\alpha_z t_\psi{}^2 \right) \right|; \ t_\psi = \omega_z/\alpha_z;$$

$W_C = W_A$: Half-width of Oriented Kinematic Collision Box of A.

$L_C = L_A$: Half-length of Oriented Kinematic Collision Box of A.

$H_C = H_A$: Half-height of Oriented Kinematic Collision Box of A.

$v_x$, $v_y$, $v_z$: Linear velocity of A corresponds with three axes X, Y, and Z.

$a_x$, $a_y$, $a_z$: Minimum linear deceleration of A corresponds with three axes X, Y, and Z.

$\omega_x, \omega_y, \omega_z$: Angular velocity of A corresponds with three axes X, Y, and Z.

$\alpha_y$, $\alpha_y$, $\alpha_y$: Angular deceleration of A corresponds with three axes X, Y, and Z.

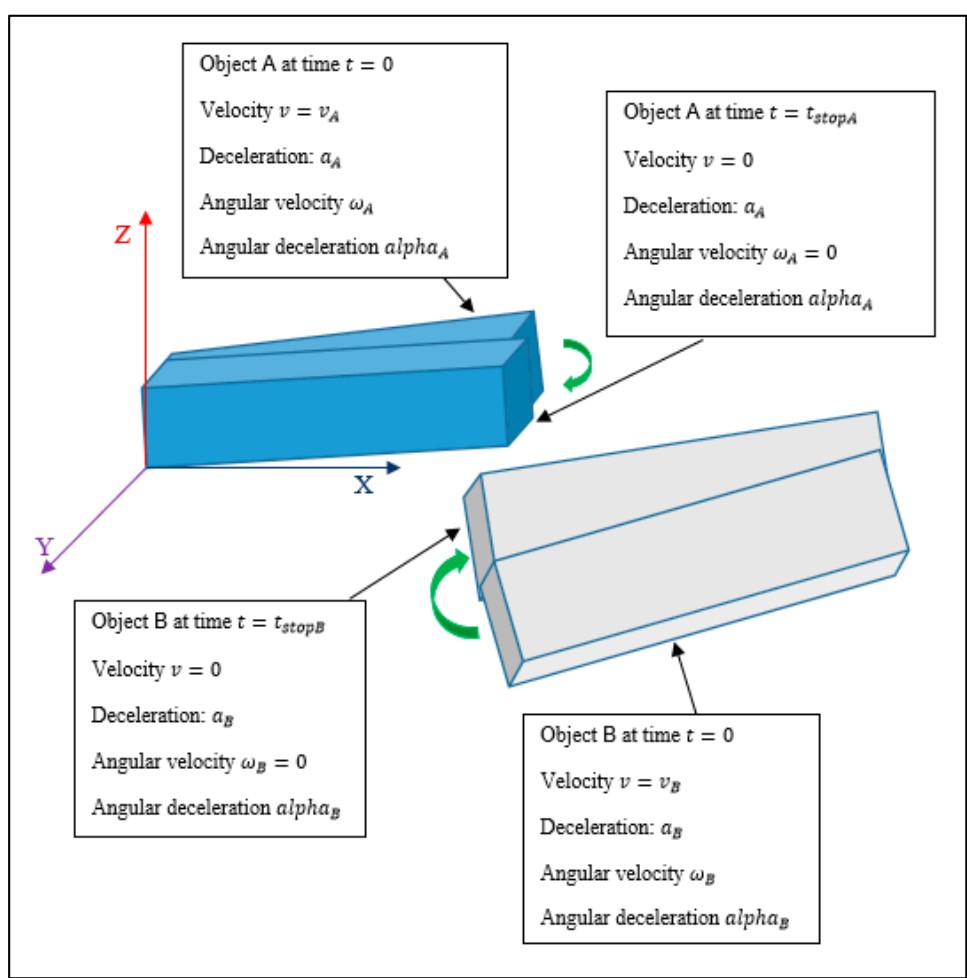

**Figure 8.** Oriented anti-collision box of objects.

## 4. Simulation

### 4.1. Drilling Floor Layout in the Simulation

Even though drilling rigs/platforms have a variety of designs depending on the location, functions, working conditions, the drilling floor area normally has similar equipment and arrangements to conduct the principle tasks of the drilling process, such as: pipe-connecting, tripping, stand-building, etc. [22].

In this paper, the layout of the drilling floor simulates a practical one, which is designed for a drill ship [22]. Figure 9 represents the overall layout of the drilling floor on a ship, where the drilling platform and derrick located at the middle of the deck. The drilling floor is constructed to perform the stand-building independent of the rotating process with dual HydraRacker and HydraTong system. The visualization of equipment on the drilling floor is constructed in Unreal Engine and described in Figure 10, and it includes the following main components:

a.    Draw-works/Top Drive
b.    Mainwell HydraTong (MW_HT)
c.    Auxiliary Well HydraTong (AW_HT)
d.    Main well HydraRacker (MW_HR)
e.    Auxiliary well HydraRacker (AW_HR)
f.    Catwalk System (CW)
g.    Main well
h.    Auxiliary well (Mouse Hole)

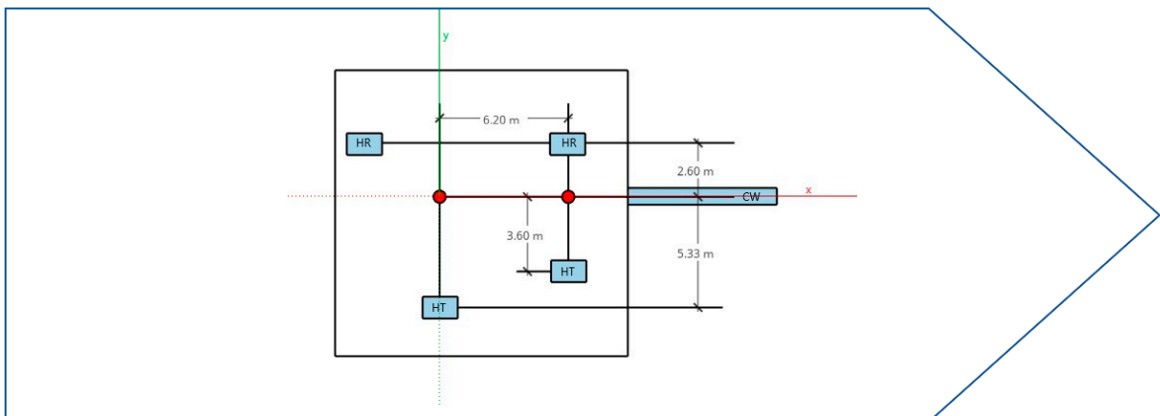

**Figure 9.** The layout of the drilling floor.

Each piece of machinery in the system is acquired with sensors to determine their position in the drilling floor and these data are transfer to the server. Real-time parameters concerning the movement of equipment, such as direction, velocity, acceleration, and deceleration are recorded and updated to the database when change occurs.

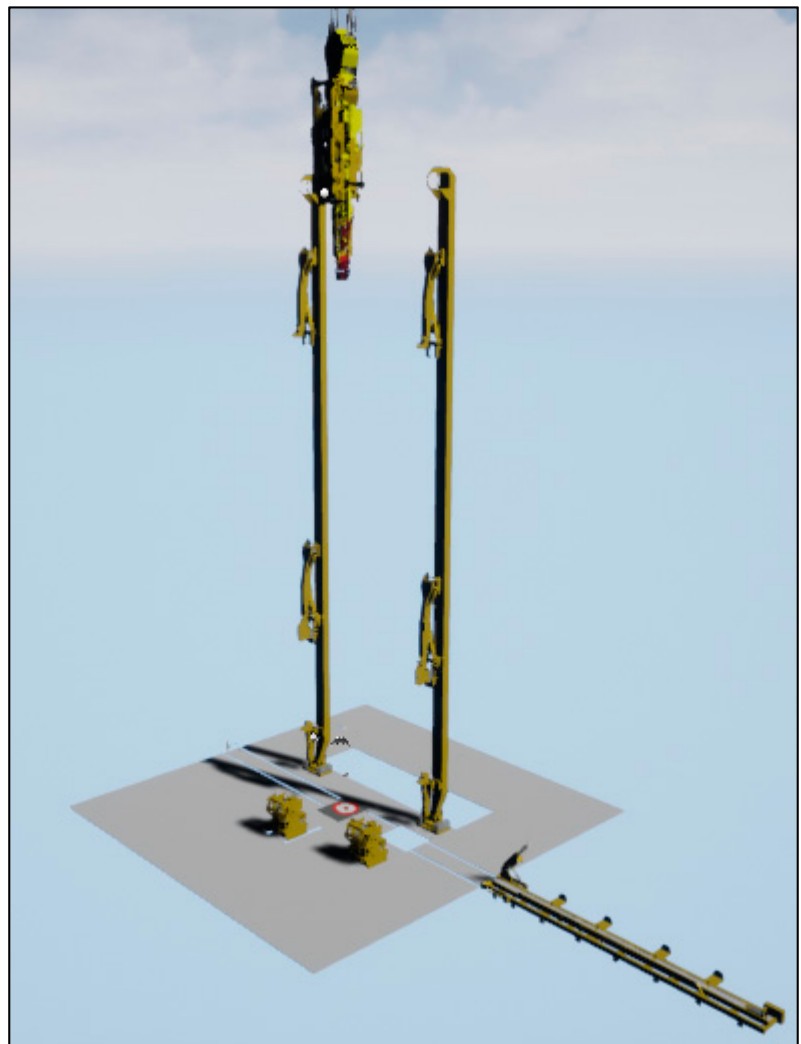

**Figure 10.** The visualization of the drilling floor in Unreal Engine 4.

*4.2. Simulation Scenarios*

To ensure the validation of the kinematic collision box algorithm, a simulating environment is established, which is capable of controlling the machinery on the drilling floor to perform all of the tasks associated with the drilling process. All dynamic characteristics and kinematic parameters of the drilling equipment, including position, velocity, acceleration rate, and deceleration rate are constructed exactly in the simulation tool. Therefore, the results acquired from the simulation provide a reliable mean of checking the performance of ACS.

The functionality of the kinematic collision box algorithm was verified by analyzing its simulation results for different scenarios, and this process was designed to emulate possible interferences of equipment during the drilling operation.

**Scenario 1:** When tripping out occurs, the top drive is lowered while the HydraTong is still in the Main Well pipe disconnecting area.

**Scenario 2:** In the process of performing a dual-task: Stand-building in the Auxiliary Well and normal drilling process in the Main Well, two HydraRacker Trolleys collide.

**Scenario 3:** During Stand-building, Auxiliary Well HydraTong and HydraRacker Lower Arm interfere while making drill pipe connection.

For each scenario, the simulations were performed for 2 cases:

1.  Using original collision box of Unreal Engine to cohere with ACS.

2.　　　Applying kinematic collision box algorithm to ACS.

*4.3. Simulation Results*

The results from simulation of Scenario 1 are represented in Figures 11 and 12. Figure 11 shows the images from simulations of 2 cases in Scenario 1 when the top drive and the HydraTong collide. In Scenario 1, the top drive starts to move from the top of the derrick with an initial velocity of 0, which increases with time until it reaches the maximum speed. The kinematic collision box algorithm has functionality to calculate the position, size, and orientation of the collision box from the bounding box of the top drive, which allows ACS to identify a possible collision earlier at about 9.4 s. After ACS sends information about the status of the collision to the machine's PLC, the top drive is stopped at about 11.7 s. Figure 12 illustrates the functionality of kinematic collision box algorithm in preventing the collision between the top drive and the HydraTong. Without the kinematic collision box algorithm, ACS detects possible interference by UE4 original collision cox at 10.5 s, but the top drive and the MW HydraTong already have collided. Similar results can be found by analyzing the simulation results of Scenario 2 and Scenario 3. Figure 13 shows the images from simulations of 2 cases in Scenario 2 when the two HydraRackers collide. The variations in the velocity and distance from the MW_HydraRacker to AW_HydraRacker are represented in Figure 14. Figure 15 shows the results of Scenario 3, when kinematic collision box algorithm is used to prevent the collision between the HydraTong and Hydraracker's Lower Arm.

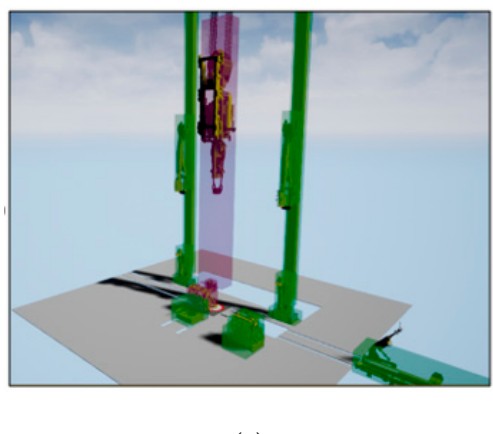 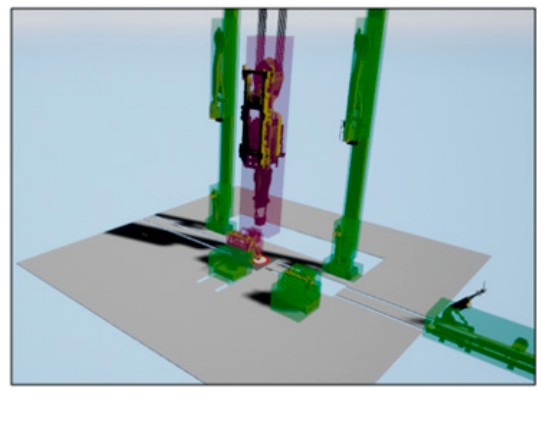

(**a**)　　　　　　　　　　　　　　　　　　　　　　　　　(**b**)

**Figure 11.** Results of the simulation in 2 cases of Scenario 1: (**a**) Kinematic Collision Box Algorithm (**b**) Original Collision Box.

In addition, another approach that can be used to determine the collision box is to use a collision box that is larger than the original bounding box and has a constant size. The extended proportion of the collision box is defined by the distance from the position where object receive the stop signal to the position that the velocity of object is 0. An extra simulating case of Scenario 1 was constructed to compare the performance of this approach and the kinematic collision box algorithm. In this case, the size of the collision box of the top drive was extended in all dimensions with an additional length of 4.92 m and the top drive starts to move down at a height of 22 m corresponding to the drilling floor.

However, this method gives inaccurate information concerning the detection of a collision when the speed of the top drive is increasing but has not reached its maximum velocity. Figure 16 shows that with the extended bounding box of the top drive, ACS triggered the collision earlier than when the kinematic collision box algorithm was used. When the top drive stops after ACS sent a signal to the PLC, the distance between the top drive and the HydraTong is still 2.67 m. Figure 17 illustrates the differences in the scale of the collision box between using extended bounding box and kinematic collision box algorithm.

The results of the simulation indicated that the kinematic collision box algorithm allowed ACS to trigger prematurely concerning a possible collision, which is enough to stop the equipment with their deceleration rate. On the other hand, ACS with the UE4 original collision box sent signals of interference only when the collision had occurred. Several additional scenarios were simulated, and the kinematic collision box algorithm demonstrated the functionality and reliability when associated with ACS to prevent the interference of machinery on the drilling floor.

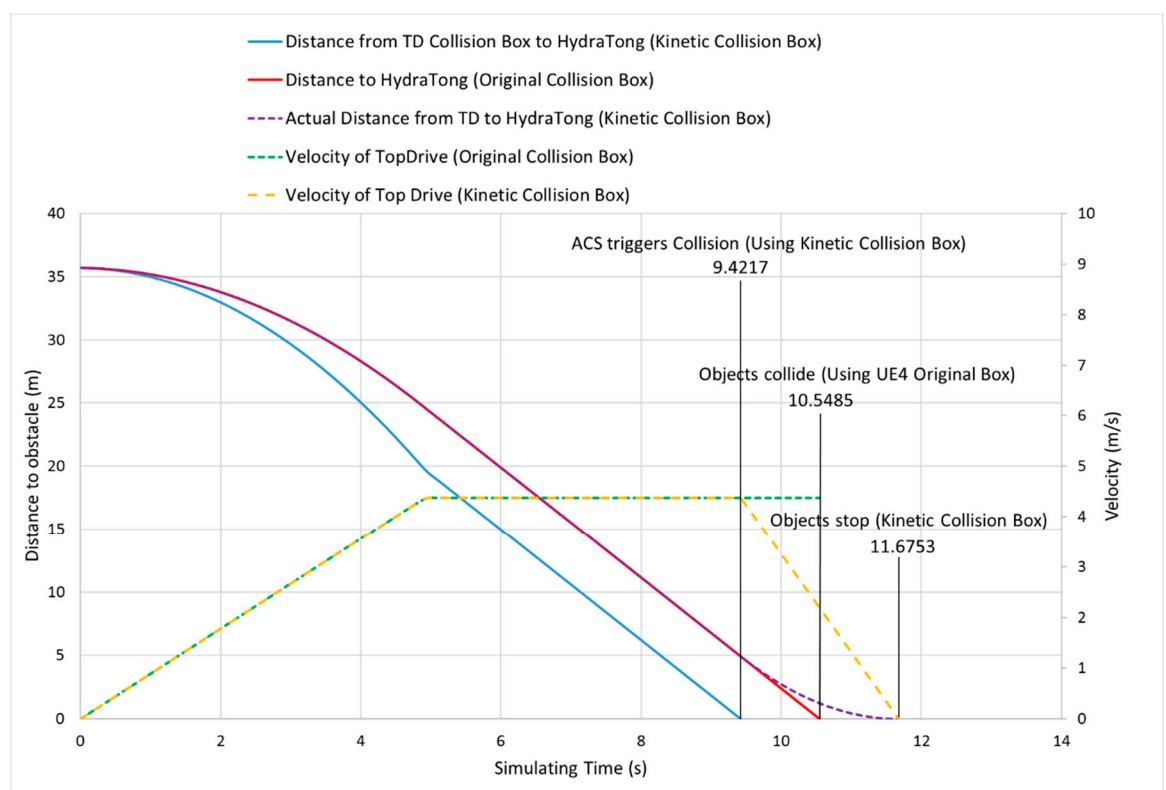

**Figure 12.** Functionality of the kinematic collision box algorithm in Scenario 1.

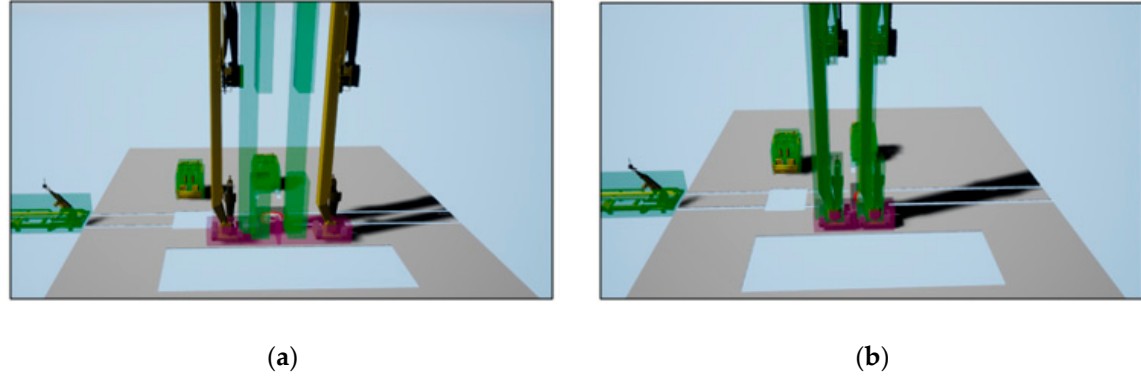

(**a**)                                                                 (**b**)

**Figure 13.** Results of the simulation in 2 cases of Scenario 2: (**a**) Kinematic Collision Box Algorithm (**b**) Original Collision Box.

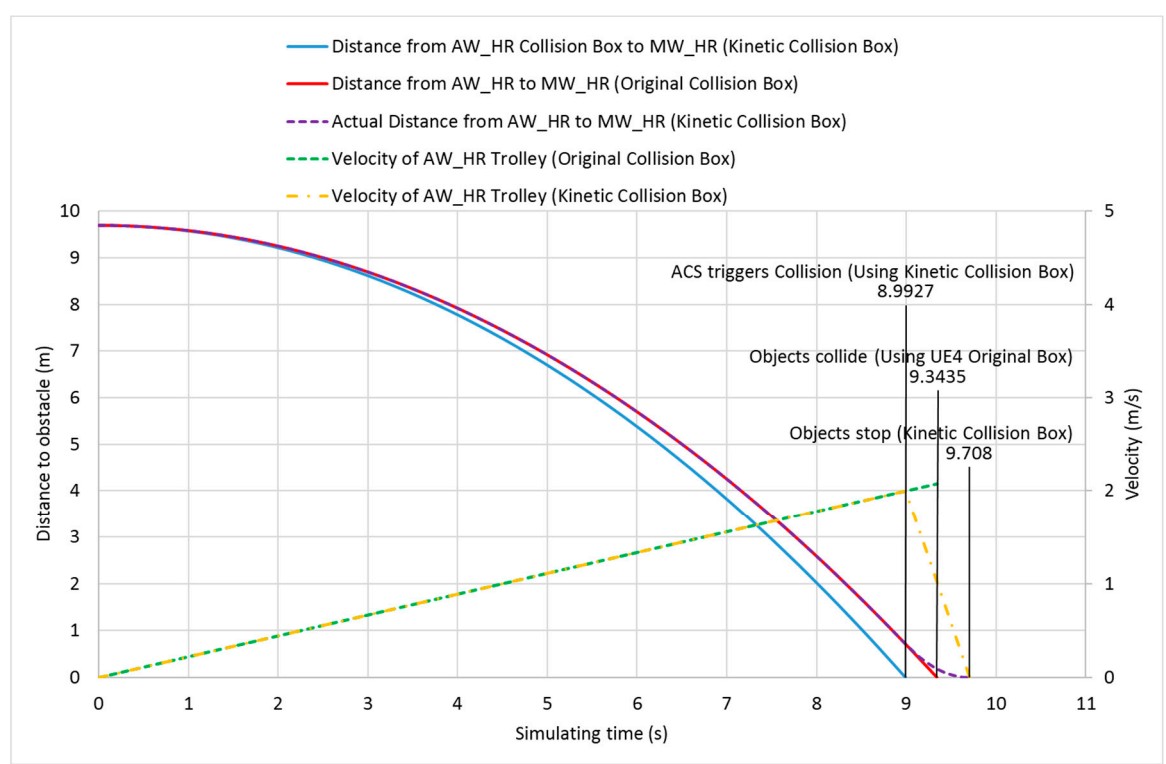

**Figure 14.** Functionality of the kinematic collision box algorithm in Scenario 2.

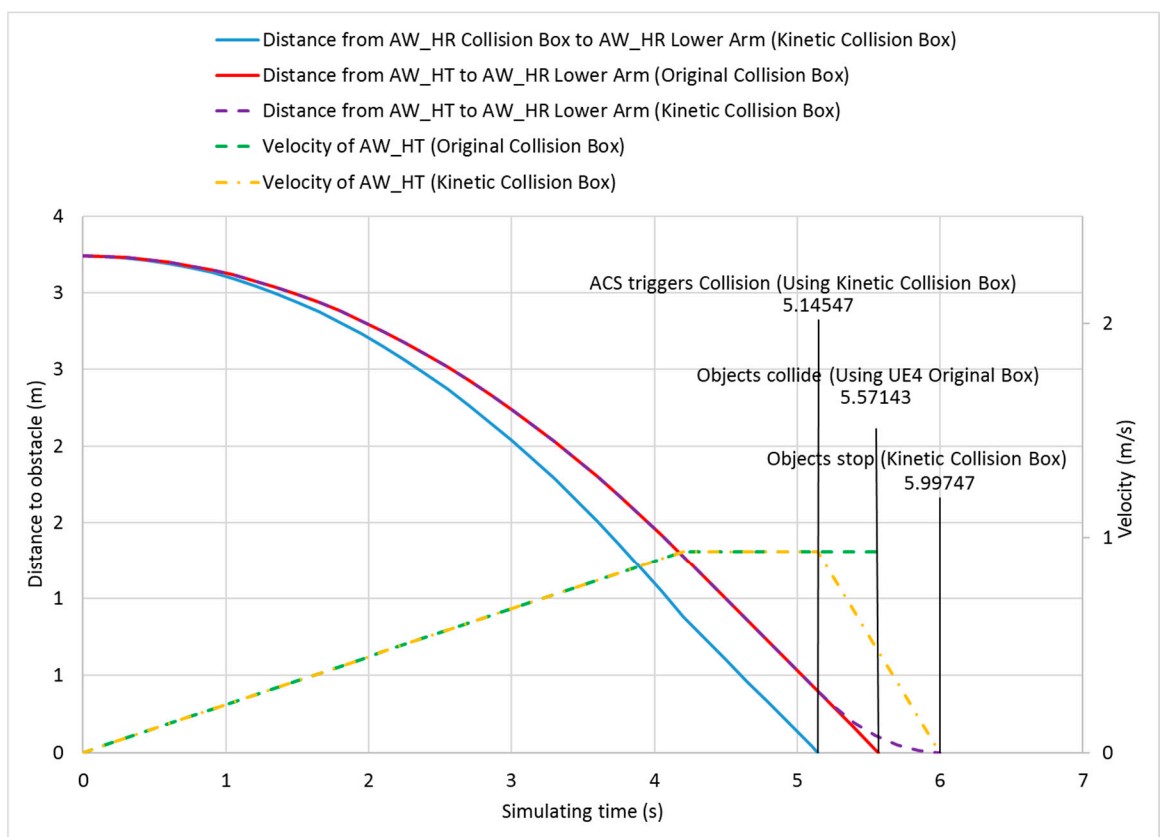

**Figure 15.** Functionality of kinematic collision box algorithm in Scenario 3.

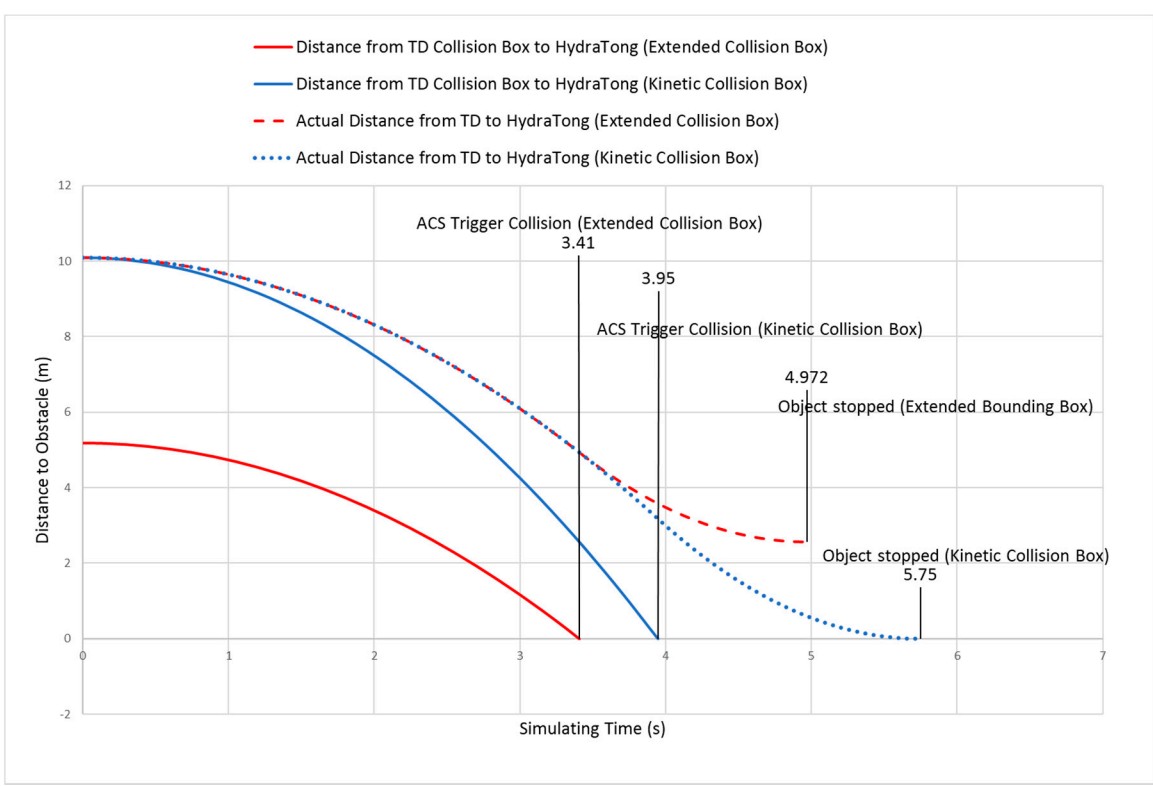

**Figure 16.** Comparison of the functionalities of the kinematic collision box and the Extended Bounding Box.

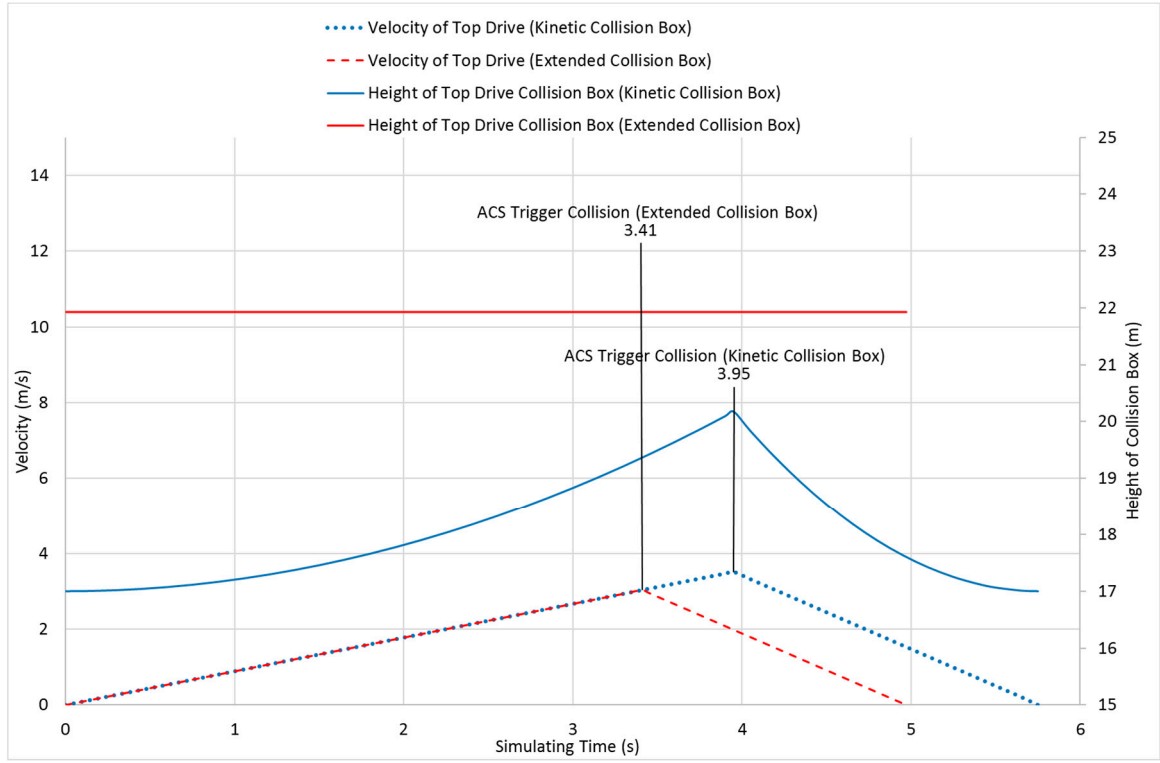

**Figure 17.** Scale of the Collision Box using the kinematic collision box algorithm and the extended collision box.

## 5. Conclusions

By analyzing the motions and using kinematic parameters of drilling equipment arranged on an offshore drilling vessel, the kinematic collision box algorithm was developed in order to provide efficient means of collision detection between equipment on drilling floor. While the conventional collision-avoidance algorithm uses collision boxes with fixed size, the proposed kinematic collision box algorithm uses collision boxes with flexible considering the movement of equipment such as velocity and deceleration rate. In additions, several simulation scenarios had been conducted to demonstrate the advances of the kinematic collision box algorithm compared to the conventional collision algorithm. Average execution time to return the collision check result was less than 10 ms, which is fast enough considering that control commands are made every 100 ms or 200 ms.

Via cohering up-to-date technology with current drilling system, the kinematic collision box algorithm and the ACS proved the functionality to be a reliable auxiliary safety system in the drilling operation. The algorithm itself and the ACS proposed an effective approach to avoid safety incidents, downtime, and operational inefficiencies due to the hazards of machine interference. Additional developments can be achieved by using artificial intelligence (AI) to optimize the algorithm or by performing the collision detection sequence instead of using the attached sequence in the current software (Unreal Engine 4). Machine Learning is a possible choice to incorporate with the algorithm and other graphic tools to increase the efficiency of collision detection between equipment on the drill floor.

**Author Contributions:** Conceptualization, J.L.; methodology, N.K. and J.L.; supervision, K.H.J.; writing—original draft preparation, D.T.N.; visualization, K.-Y.K.; writing—review and editing, J.L. All authors have read and agreed to the published version of the manuscript.

**Funding:** This research was supported by the Industrial Technology Innovation Program (grant no. 10070163, Development of Integrated Control System and HILS based Verification System about Offshore Drilling Rig) and PNU Korea-UK Global Graduate Program in Offshore Engineering (N0001288) both funded by the ministry of trade, industry & energy of Korea.

**Conflicts of Interest:** The authors declare no conflict of interest.

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
