# Peer review of "A Kinematic Collision Box Algorithm Applied for the Anti-Collision System of Offshore Drilling Vessels"

_jmse, doi:10.3390/jmse8060420_

Round 1
Reviewer 1 Report
The presented algorithm is based on the motion vector, velocity and acceleration of the object.
Although the background is clearly represented in the mathematical equation, there are some errors.
First, the equation on lines 236 and 260 seems identical. There are three different time values associated with the x, y, z axis, and another three indexed times defined with respect to angular velocities. What is the difference between the six indexed times defined? In the results of the simulation, as in Fig. 7, there is only one time.
To Fig. 8 would be clearer if the axes were signed and connected to the objects.
The conclusions could be more extensive.
Reviewer 2 Report
- The paper title does not correspond with the content. The title indicates the algorithm will be mainly presented and analyzed: "A Collision Box Kinematic Algorithm Applied for the 2 Anti-Collision System of Offshore Drilling Vessels", while the majority of the paper concerns the whole system.
- Introduction: There is no information on the state of the art of the problem which is solved by the algorithm. There is no exact literature review also. There is some sort of information in "3.1. Conventional Collision Box" but it is not enough.
- 3. 1.1. Collision detection: There is no specific justification why just this and not another method was selected: "The most common method is based on the Bounding Box Volume Hierarchies. The major reason
for the popularity of the Bounding Box Volume Hierarchies method is that it uses a bounding volume that usually is the simplest geometric primitive that encloses one or more objects to conduct 154
overlapping tests. Thus, it is considered to be a cheaper test, but still has high efficiency when compared with other methods". Why other methods are not proper. What kind of disadvantages do they have? - What kind of novelty does the solution have?
- What about the execution time of the algorithm? Is it fast enough to implement it in a real-life system?
- Conclusions: "Additional developments can be achieved by utilizing Artificial Intelligent (AI) to optimize the algorithm or by performing the collision detecting sequence instead of using the attached sequence in the current software (Unreal Engine 4)" What is the exact reason of the AI application? What kind of AI methods do the Authors plan to use? What about plans for solution implementation in real-life systems?
Author Response
Please see the attachment. Thank you for your insightful review.

Reviewer 3 Report
The authors have prepared a well-written and presented manuscript on a topic of good importance. The authors elaborated sufficiently on all details of the anti-collision systems and show good results of the application.
The only point which may require some additional comments by the authors is how and to what extent the system they propose improves the current state-of-the-art.
Author Response
Thank you for your comment. As mentioned in the paper, ACS logic in the current drilling industry is still primitive in checking possible collisions. We expect that our algorithm is the starting point to extend the logic based on behavioral analysis. Please refer to the revised conclusion. After some operational experience with other systems composing the integrated drilling system, we are going to implement a machine learning algorithm into the ACS. Not only the current status of the machine, but also the pattern of operators' behavior will be analyzed and ACS will be the core of the whole safety assurance system.
Round 2
Reviewer 2 Report
It's ok